# Sex Differences in Work-Stress Memory Bias and Stress Hormones

**DOI:** 10.3390/brainsci10070432

**Published:** 2020-07-08

**Authors:** Laurence Dumont, Marie-France Marin, Sonia J. Lupien, Robert-Paul Juster

**Affiliations:** 1Centre de recherche de l’Institut Universitaire en Santé Mentale de Montréal, Montreal, QC H1N 3V2, Canada; marin.marie-france@uqam.ca (M.-F.M.); sonia.lupien@umontreal.ca (S.J.L.); robert-paul.juster@umontreal.ca (R.-P.J.); 2Department of Psychiatry and Addiction, Université de Montréal, Montreal, QC H1N 3V2, Canada; 3Center for Studies on Human Stress, Montreal, QC H1N 3V2, Canada; 4Department of Psychology, Université du Québec à Montréal, Montreal, QC H3C 3P8, Canada; 5Center on Sex*Gender, Allostasis, and Resilience, Montreal, QC H1N 3V2, Canada

**Keywords:** cortisol, declarative memory, workplace stress, sex differences

## Abstract

Mental health problems related to chronic stress in workers appear to be sex-specific. Psychosocial factors related to work–life balance partly explain these sex differences. In addition, physiological markers of stress can provide critical information on the mechanisms explaining how chronic stress gets “under the skull” to increase vulnerability to mental health disorders in working men and women. Stress hormones access the brain and modulate attentional and memory process in favor of threatening information. In the present study, we tested whether male and female workers present a memory bias towards work-stress related information, and whether this bias is associated with concentrations of stress hormones in reactivity to a laboratory stressor (reactive levels) and samples taken in participants’ workday (diurnal levels). In total, 201 participants (144 women) aged between 18 and 72 years underwent immediate and delayed recall tasks with a 24-word list, split as a function of valence (work-stress, positive, neutral). Participants were exposed to a psychosocial stressor in between recalls. Reactivity to stress was measured with saliva samples before and after the stressor. Diurnal cortisol was also measured with five saliva samples a day, during 2 workdays. Our exploratory results showed that men presented greater cortisol reactivity to stress than women, while women recalled more positive and neutral words than men. No sex difference was detected on the recall of work-stress words, before or after exposure to stress. These results do not support the hypothesis of a sex-specific cognitive bias as an explanatory factor for sex differences in stress-related mental health disorders in healthy male and female workers. However, it is possible that such a work-stress bias is present in individuals who have developed a mental-health disorder related to workplace stress or who have had one in the recent past. Consequently, future studies could use our stress memory bias task to assess this and other hypotheses in diverse working populations.

## 1. Introduction

Women have been increasing and sustaining their presence in the workforce for more than half a century [1,2]. This, along with changes in economic and socio-political pressures has led to reorganization of workplaces around the world and increasing work overload for both sexes [3,4,5]. Although greater equality between men and women in the workplace is slowly being achieved, research shows that mental health problems related to stress at work are still sex-specific. As an example, women are approximately twice as much at risk than men to experience depression [6,7]. On the other hand, men are between 1.5 and 3 times more likely to experience alcohol-related problems than women [7,8,9].

There are two leading psychological models to explain the levels of stress experienced in a workplace setting that have been successful at identifying which work-related factors can explain psychological distress [10,11]. The first is the Demand-Control-Support model [12,13,14] and the second is the Effort-Reward Imbalance model [15,16]. To date, however, psychological models of workplace stress do not usually take into account physiological markers of stress, which could help explain discrepancies in the health consequences of workplace stress in men and women.

### 1.1. Biological Models of Stress

Animals and humans respond to stress by activating the hypothalamic-pituitary-adrenal (HPA) axis. This activation unfolds by first perceiving a stressor, which triggers the hypothalamus to secrete corticotropin releasing factor. This, in turn, leads to the release of adrenocorticotropic hormone by the pituitary gland, which finally signals the adrenal glands to release glucocorticoids (cortisone in animals and cortisol in humans). Cortisol helps to mobilize energy through glucose metabolism at the expense of other systems (such as reproduction, immunity, inflammation and growth) which are not immediately required to face acute stressors [17]. Without the overt presence of stressors, diurnal cortisol levels follow a 24 h circadian profile, with an AM peak and a PM trough. In this manner, cortisol concentrations peak in the morning and slowly decline during the afternoon, evening and night. It has been established in animal studies that activating the HPA axis beyond the usual basal levels is only adaptive when this activation is brief, proportional to the magnitude of the stressor, and when baseline levels are promptly recovered [18].

On the other hand, long lasting stressors can lead to a state of chronic stress that can be detrimental to both physical and mental health. Cortisol can cross the blood–brain barrier and bind to glucocorticoid receptors mostly concentrated in the prefrontal cortex, the amygdala, and the hippocampal formation [19]. These structures also play a role in regulating the HPA axis, which can lead to a self-sustaining chronic stress state [20]. In both children and adults, cortisol in the brain has the potential to affect memory [21,22,23,24,25] and emotional regulation [26,27,28,29,30].

The brain structures affected by cortisol also play a role in identifying and interpreting situations as being stressful (or threatening), and in the selection and/or inhibition of potential responses to these situations [31]. Chronic dysregulation in glucocorticoid levels has been associated with increased risk for depression and burnout, the two conditions showing respectively increased [32] or decreased [33] levels of cortisol. Considering biological markers of stress can therefore provide crucial complementary information to understand how chronic stress can get ‘under the skull’ to increase vulnerability to different mental health disorders in the workplace in men and women.

Although most biological studies reveal no sex differences in diurnal cortisol levels, most report the presence of an important sex difference in cortisol reactivity to a laboratory stressor [34]. Indeed, studies show that men present significantly greater cortisol reactivity to a stressor when compared to women [35,36,37]. Initially, sex hormones were proposed as an explanatory factor for this difference and this mechanism has received some empirical support [35,38]. However, the administration of medication suppressing the release of sex hormones before exposure to a laboratory stressor does not completely eliminate the aforementioned sex differences [39], pointing towards different or complemental mechanisms of action that need to be explored further.

### 1.2. Cognition and Stress at Work

One of the mechanisms explaining the discrepancy in prevalence of work-related mental health disorders could be differences in cognitive processing of information as being stressful or threatening in men and women. More precisely, differences in attentional bias towards stress-related information may be sex-specific, and this could drive sex differences currently observed in stress responses. Attentional circuits preferentially detect and process information in the environment that has immediate survival value and task relevance through selective attention [40,41,42]. The individual parameters that determine relevance of stimuli (as threatening or not) and adequacy of response are largely determined by the individual’s previous experiences throughout life that impact the individual’s perception of stressors [43,44].

Cognitive systems are mostly shaped in childhood during critical developmental periods. Notwithstanding, but experiences later in life can also have significant influence in higher-order cognitive and emotional processes. For example, children who have been maltreated show marked sensitivity in detection of anger-related content [45,46,47]. Adults with anxious symptoms and/or anxiety disorders show preferential biases towards information that is specific to the feared foci of their anxiety [48,49]. It is important to note that studies performed in human populations reveal that this attentional bias to threatening information is partly sustained by the secretion of glucocorticoids that are released during a stressful and/or an emotional experience [26,50,51].

As previously mentioned, secretion of glucocorticoids can be adaptative. In that sense, the enhancement of memory for stimuli inducing stressful and/or emotional responses may be essential for species’ survival. However, in a context where survival is not at stake and stressful stimuli are present in a repeated way (such as in some workplace settings), chronic activation of the HPA axis might lead to an increased encoding and/or consolidation of stress-related information (bottom-up effects). This attentional bias may then lead to increased secretion of stress hormones (top-down effects). This, in turn, leads to a vicious circle in which the stress hormones released in response to a chronic stressful condition access the brain and modify the way that upcoming information are perceived and interpreted, thus leading to increased stress reactivity [52,53].

Without underestimating the complexity of the human brain, the objective of the current study was to (1) identify if this mechanism, starting from the “filtering” mechanism can lead to a bias that is specific to workplace stress (which would modulate memory performance in response to stress), (2) if this cognitive mechanism is similar in healthy male and female workers, and (3) if it relates to biological markers of stress. Given the lack of previous data on this particular issue and the empirical nature of the protocol, we did not propose any hypothesis with regards to directionality of significant effects.

### 1.3. Rationale and Historical Background of the Present Study

From 2011 to 2015, our laboratory conducted a study among healthy male and female workers in order to perform a Sex (differences between men and women related to biological factors) by Gender (differences between men and women related to psychosocial factors such as gender roles and socio-cultural factors) analysis on psychological and physiological markers of stress. Various papers have been published on the results of this confirmatory study [54,55,56]. While we were preparing the research protocol for this study, and based on data from our laboratory showing the presence of stress-induced attentional bias in young adults as a function of childhood socioeconomic status [51], we decided to develop a new cognitive task that would allow us to assess whether male and female workers differ on the recall of information related to (1) stress at work (2) positive information or (3) neutral information. The goal of this new *Work-Stress Memory Task* (WSMT) was to determine if a potential memory bias toward work-stress related information is present in male and female workers and whether it is associated with biological markers of stress. This paper presents the results of this exploratory study. Our purpose was to assess the WSMT performance before and after exposure to a laboratory stressor in male and female workers. The rationale and pre-analysis plan of this exploratory project has been pre-registered and can be found at OSF under the following address: https://osf.io/tcbuh.

## 2. Materials and Methods

### 2.1. Sample and Missing Data

From an initial pool of 295 participants recruited among employees of the Institut Universitaire en Santé Mentale de Montréal (IUSMM) hospital, a total sample of 204 working adults completed the original study between 2011 and 2015. The IUSMM is the largest psychiatric hospital in the Canadian province of Québec and, at the time, had a total of 1546 employees (65% women) from different professions. Participants in the study came from clinical services (29.9%), administration (17.2%), research (13.7%), social integration (11.3%), professional services (9.8%), maintenance (10.8%), general direction (4.4%) and human resources (3.0%). Participants were allowed to complete the study during their normal work hours. This study (including the WSMT) was approved by the Research Ethics Board of the same institution (2011-003).

For the current set of analyses, 201 of the participants were included since the WSMT could not be completed for technical reasons for 3 out of the 204 participants. Women comprised 70% of the sample (*n* = 141), and participant ages ranged from 18 to 72 (Average—40.48 years old; Std Dev—12.17).

### 2.2. Questionnaires and Tasks

#### 2.2.1. Socio-Demographic and Psychosocial Information

In our analyses, we used the age (ranging from 18 to 72) as a covariate and biological sex (man/woman) of participants (all participants were cisgender) as a between-subject factor. Education level (number of years of school completed), body mass index (BMI) and self-reported chronic stress were also compared between men and women for descriptive purposes. Self-reported chronic stress was measured using the Trier Inventory for Chronic Stress (TICS), a 30-item questionnaire with 10 subscales that asks the participant to rate the levels of stress they experienced in the past month, mostly in work-related circumstances [57]. Potential differences in socio-demographic information, BMI [58] and self-reported chronic stress [59] were verified in order to make sure that men and women sub-samples were comparable.

#### 2.2.2. The Work-Stress Memory Task—WSMT

The WSMT is an emotional declarative memory task developed by the Center for Studies on Human Stress [60]. The task comprises an initial encoding phase in which a list of 24 words is presented visually using E-Prime2 (Psychology Software Tools, Pittsburg, PA, USA). Each word is presented once for five seconds, in a random order, with explicit instructions to remember the words. This encoding phase is followed by an immediate recall of the list, in which participants have to write down all the words they remember within one minute. A delayed recall phase is conducted approximately 30 min later (after exposure to a psychosocial stress task), both recall phases are completed using pen and paper.

The 24-words list has 3 different types of word: Work-stress, Positive and Neutral. Eight words are presented in each of these three categories. In order to develop the list of work-stress words, we asked 208 working adults attending stress management workshops and conferences in different organizations to write down words that “best described stress in the workplace for them”. Note that none of the participants in the original sample of this study were part of this initiative. Individuals were mostly working in IT/Engineering (30.8%), management or direction (30.8%), administrative support (8.2%) and human resources (7.2%). A frequency analysis was then performed on the answers provided by participants and we extracted the 8 most frequent words provided by the group. Positive and Neutral words were matched to work-stress words for length, and frequency of use in the English language. We also made sure that the meaning of the positive and neutral words was not directly related to work, and they were obtained using standardized word banks. The words of the WSMT can be found in Table A1 of Appendix A.

Performance on the WSMT was calculated using the number of words recalled for each category. A perfect recall of all words of a category gives a score of 8 and recalling none of the words of a given category gives a score of 0.

#### 2.2.3. Trier Social Stress Task (TSST)

The TSST is a laboratory-based stressor developed by Kirschbaum and colleagues [61,62] and is one of the standard stress interventions used in a laboratory setting aimed at provoking activation of the HPA axis. It has two main phases. The first is an anticipation phase in which participants are given the instructions of the task and asked to prepare themselves. The second phase is the performance phase, in which a panel of “experts” stands on the other side of a one-sided mirror (in the panel-in version of the TSST), observing and correcting the participant during a speech and an arithmetic task. Samples of saliva are taken before and after completion of the TSST to assess cortisol reactivity to stress.

### 2.3. Protocol

Participation in this project consisted of two laboratory visits, two days of saliva samples at work and home, one day of saliva samples during a rest day, home questionnaires and a follow-up call. During the first laboratory visit, participants gave informed consent, completed the WSMT, the TSST and some questionnaires, while providing saliva samples every 10 min for a total of 6 samples aimed at measuring cortisol levels. At-home saliva samples followed consensus guidelines [63]: samples were taken upon awakening, 30 min later, at 2 PM, at 4 PM and upon going to bed. Participants recorded the time they took their samples and a Medication Event Monitoring System (MEMS^TM^, AARDEX Ltd., Sion, Switzerland) was used to optimize compliance [63]. Questionnaires were also completed at home on a secure online platform. In this paper, results to the Trier Inventory for Chronic stress will be reported, as previously mentioned. The full list of questionnaires in the original protocol can be found in Appendix A
Table A2. Note that questionnaire instructions and practice were provided to familiarize participants with our platform; however, participants completed questionnaires at home at their leisure. During their second laboratory visit, participants handed their saliva sample and a blood sample was taken. A full list of biological measures that have been collected in this protocol can be found in previous publications using this sample [54,55,56] and in our pre-registration plan (see above).

#### Hormonal Measures

To obtain assays, frozen samples were first brought to room temperature, then centrifuged at 150× *g* (3000 rpm) for 15 min. High-Sensitivity immunoenzyme assays were used for cortisol (Salimetrics^®^, No. 1-3102). These have sensitivity of 0.012–3 μg/dL, with inter- and intra-assay coefficients of variance, respectively, below 9.27% and 5.89%. For each sample, assays were run in duplicate, then averaged.

To assess the diurnal levels of stress hormones at work and home, the five samples from the two work days were considered. As we were interested in work-stress, the weekend samples were not considered in these analyses. The cortisol concentrations at each time point were averaged for the two days and the area under the curve (from the ground—AUCg) was calculated using Pruessner and colleagues’ method [64]. To assess reactive levels of stress hormones, the six samples taken during the laboratory visit described in the previous section were used. We also calculated the area under the curve for these values, but using the increase from baseline formula (AUCi) [64].

### 2.4. Statistical Analysis

Statistical analyses were conducted using IBM Statistical Package for the Social Sciences 25. Although our research objectives were pre-registered, our analyses deviate from the initial plan as we now concentrate our analyses on sex differences. For the present study, preliminary analysis assessing sex differences in socio-demographic information and hormonal measures were first conducted using independent sample t-tests and repeated measures factorial ANCOVA to set the grounds and compare the current sample to the existing literature. Results for the WSMT were then analyzed using a repeated-measures factorial ANCOVA, comparing performance at the immediate and delayed recall, for each word type, for men and women. This was done while controlling for the effects of age, given its well-established effects on declarative memory [65,66]. Finally, performance on the immediate and delayed recall of the WSMT, for each type of words, and for men and women, was correlated with diurnal cortisol AUCg and reactive cortisol AUCi using Pearsons correlations.

## 3. Results

### 3.1. Preliminary Analysis

As shown in Table 1, independent sample t-tests revealed no sex differences in age, education level, number of hours worked, BMI and self-reported chronic stress (TICS) levels.

Further preliminary analyses were conducted with repeated measures factorial ANCOVA to replicate previously established time and sex differences in TSST reactivity and absence of sex differences in diurnal cortisol levels, while controlling for age of participants, as shown in Figure 1. When comparing cortisol samples collected over the TSST session, there were expected effects of time (F(5, 995) = 4.313; *p* = 0.001) and sex (F(1, 199) = 6.092; *p* = 0.014), but no interaction effect (F(5, 995) = 2.065; *p* = 0.068). Cortisol levels significantly increased starting at the TSST, peaked just after and started decreasing shortly thereafter. Cortisol levels in men were significantly higher than those of women.

In contrast, statistical analysis of basal cortisol values revealed a time effect (F(4, 740) = 46.887; *p* < 0.001), but no sex (F(1, 185) < 0.001; *p* > 0.999) or interaction (F(4, 740) = 0.803; *p* = 0.524) effects. Both men and women showed a typical cortisol diurnal pattern, peaking 30 min after awakening.

### 3.2. Performance on the Work-Stress Memory Task

Performance on the WSMT for the immediate and delayed recall phases is presented in Figure 2 and results of the repeated measures factorial ANCOVA, comparing immediate and delayed recall of all type of words, for men and women, are shown in Table 2. Main effects of recall phase and sex were detected, along with a significant interaction between type of word and sex. There was no main effect of the type of word and the two other interaction terms, between recall phase and type of word, between recall phase and sex, and between recall phase, type of word and sex were also not significant.

#### 3.2.1. Manipulation Checks for Performance on the Work-Stress Memory Task

Some of the significant effects in of the ANCOVA were expected in the context of a declarative memory task. Results of the main effects of recall phase and sex showed that performance on the immediate recall was systematically better than for the delayed recall, and that women recalled significantly more words regardless of the recall phase. It is also important to highlight the significant contribution of age as a covariate in this model, where memory performance was systematically lower as age increased (F(1, 198) = 23.268; *p* < 0.001).

#### 3.2.2. Sex-specific Memory Bias

Regarding effects of interest for our objectives, no significant sex-specificity could be detected regarding work-stress words. The significant interaction between sex and type of word was investigated with 95% confidence intervals and Bonferroni corrections. These comparisons show that women recalled more positive and neutral words than men, regardless of the recall phase, while controlling for age. This was not the case for work-stress words. With these same 95% confidence intervals, it was not possible to detect significant differences between word types for each sex separately.

### 3.3. WSMT Performance and Cortisol Measures

A series of Pearson correlations were conducted to determine the association between the cognitive and endocrine measures. As Table 3 shows, there were no significant correlations between memory of any type of words and either reactive or diurnal cortisol, both for immediate (all *p*s > 0.161) and delayed (all *p*s > 0.080) recall.

## 4. Discussion

The goal of this study was to determine whether male and female workers present a memory bias for work-stress words and whether this cognitive bias is associated with diurnal or reactive cortisol levels.

We first confirmed previous psychoneuroendocrine research by showing that men presented increased cortisol levels during a laboratory stressor when compared to women; however, there was no sex difference in diurnal cortisol levels. These results were manipulation checks, confirming the quality of our experimental protocol, given the robustness of these findings across the literature [35,36,37]. The second manipulation check performed on the memory task allowed us to be confident in the preliminary validity of the WSMT; namely, the presence of a decline of overall performance with age [67], higher performance in the immediate recall, and a generally better performance in women than in men [68]. These effects are also robustly found in the declarative memory literature [65,66]. While the difference between immediate and delayed recall was expected, it is not possible to know in this particular experimental design which portion of this decline in performance was attributable to the delay, versus to the deleterious effect of the TSST on memory.

### 4.1. Work-Stress Bias

In relation to our main objective, we found no preferential recall for work-stress related content, whether it be compared to other types of words, when exposed to the TSST, or when compared between men and women. This result does not support the presence of a work-stress related memory bias in one sex over the other as was originally hypothesized.

The absence of a significant memory bias for work-stress related words could be explained by the fact that the words chosen as ‘work-stress related’ may not have a consistent valence for each individual. While the way the words were chosen for the WSMT was intended to limit researcher bias, this led to the inclusion of words with potentially ambiguous valence in the work-stress category. Indeed, words such as “team” and “management” were included in the work-stress list as they were among the most frequently reported words by our control sample. However, if these words triggered a different contextual association than work-related stress during the WSMT for our participants, they could have been interpreted as neutral of positive words. This means that there may be a heterogeneous allocation of attentional resources toward these words, rather than the increased attention usually associated with negatively valanced words [42]. One way to control for this limitation in future studies would be to verify the perceived valence of the words after the second recall for each participant. This would allow to adjust the level of perceived valence and have more precise and individualized results. This methodological feature could allow to deepen our understanding of sex differences and of the interaction between sex and type of word. Identifying words that resonate with work-stress differently for men and women would make for a more fine-grained delineation in identifying a sex difference or sex-specificity that the current study could not.

Serial position of the words could also have been a confounding factor at the individual level. While randomization of the word order for each participant was meant to remove serial position effects between subjects, it probably had an effect within each individual that we have no way of statistically accounting for in the context of this study.

Also, while the TSST is a validated method to induce the activation of the HPS axis, using a stressor more closely related to situational stress as it is experienced at work could have helped prime memory biases specific to work.

One could hypothesize that the longer the exposition to workplace stress, the stronger the potential for the development of a bias. However, in the current context, experience in the workplace is closely correlated with age, which is known to be a strong predicting factor in the decline of declarative memory performance [65,66]. A more homogeneous sample in terms of age, but with varying levels of experience in a particular job could be a way to assess the effect of work experience, rather than age, on cognition and the development of cognitive biases.

Additionally, it is possible to test the presence of a work-stress memory bias in individuals who may have developed a stronger bias due to specific reasons that we did not consider. For example, individuals who are currently experiencing a mental-health disorder related to workplace stress (e.g., burnout) or who have had one in the recent past could be more susceptible to display cognitive biases in the context of work-stress. Studies have found differences in the processing of positive information in individuals recovering from a major depressive disorder [69,70] and altered working memory in women on long-term sick leave due to depression [71].

### 4.2. Sex Differences in Relation to Memory Bias and Biological Markers of Stress

Interestingly, women recalled more positive and neutral words than men, regardless of the recall phase. Although this result does not confirm the presence of a memory bias toward work-stress words, it shows the presence of a sex difference in the processing of positive and neutral content in the absence of a sex difference in the processing of work-stress content, which could be seen as an indirect bias.

It is important to note that although the absence of a sex difference in the recall of work-stress related content is interesting, it could mainly be due to a more general ‘negative memory bias’ [72]. As previously mentioned, the only potentially negative words presented to the participants in the present study were work-stress words and it is thus possible that the effects in memory are more closely related to the negative valence of the words rather than the stress-related particularity of these words. One way to disentangle this effect would be to add a category of negative (although not related to work-stress) words to the list to be memorized. Comparing the recall of work-stress and negative content could help confirm the presence of a memory bias for work-stress content in male and female workers.

It is interesting to note that women recalled significantly more positive and neutral words than men and were less reactive to a laboratory stressor when compared to men. It is thus possible that the seemingly more efficient cognitive processing of women toward positive and neutral content has a positive impact on the secretion of cortisol, leading them to produce less cortisol in response to a laboratory stressor. Although this hypothesis is interesting, one has to be reminded that we did not find any significant correlation between number of positive and neutral words recalled by women and diurnal or reactive cortisol levels. Consequently, if this effect is present, it might be too weak to be detected in our sample with the experimental design employed. Also, as previously mentioned, some studies have found differences in processing of positive information in individuals, women in particular, with previous episodes of depression [69,70,71]. However, we re-iterate the necessity for replication of these results before making stronger theoretical or practical assumptions about this unexpected result.

As we just touched upon, recall of work-stress words was not associated with diurnal cortisol levels nor to cortisol reactivity to stress. Although men reacted with greater secretion of cortisol during the TSST procedure when compared to women, we found no sex difference in recall of work-stress related content. This result does not support the hypothesis of a sex-specific cognitive bias as an explanatory factor for sex differences in stress-related mental health disorders in the workplace. Furthermore, replicating the current results with a control group without the TSST between the immediate and delayed recall could help disentangle the role of reactive cortisol and of recall phase in memory performance [73,74,75].

## 5. Conclusions

Despite the fact that our sample of healthy men and women showed typically expected cortisol profiles, both in basal and reactive conditions, and that performance on the WSMT replicated usual robust effects in other declarative memory tasks, we did not find direct evidence for a work-stress memory bias in male and female workers. However, women showed systematically better recall of positive and neutral words than men, pointing towards differential processing of non-threatening information in men and women. Memory performance was also not significantly associated with either reactive or diurnal cortisol, for both sexes. Testing the presence of such a bias in a population that may have had a stronger environmental or experiential incentive to develop it would allow us to have more definitive evidence on the preferential treatment of work-stress related information.

## Figures and Tables

**Figure 1 brainsci-10-00432-f001:**
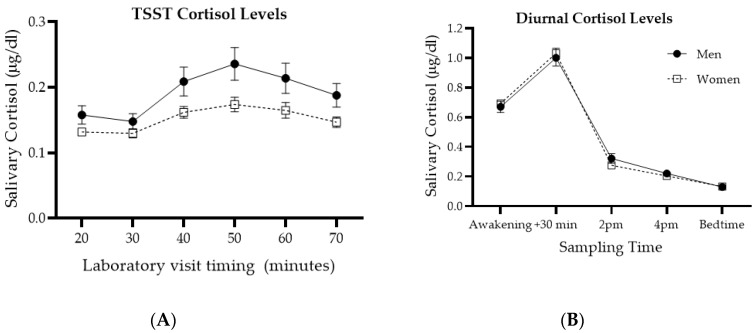
Diurnal and Trier Social Stress Task (TSST) cortisol levels for men and women. Panel (**A**) shows samples taken during the laboratory visit, each 10 min, with the TSST procedure occurring between 30 and 40 min. Panel (**B**) shows diurnal cortisol levels averaged over two typical working days. Adapted from [55]. Note that controlling for sex hormones in previous analyses unmasked a sex difference in diurnal cortisol [55].

**Figure 2 brainsci-10-00432-f002:**
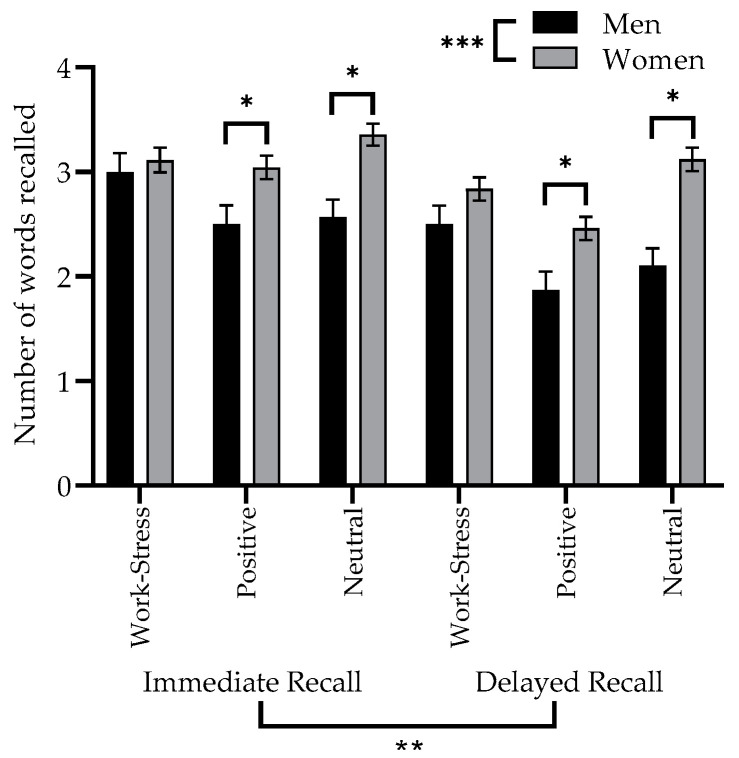
Performance on the immediate and delayed recall phases of the WSMT for each word type, by sex. Significance values indicated as follows * *p* < 0.05 or 95% CI; ** *p* < 0.01; *** *p* < 0.001.

**Table 1 brainsci-10-00432-t001:** Participant characteristics for men and women, with averages (standard deviations) and significance level of independent t-tests.

Characteristic	Men	Women	*p*
Age	38.97 (12.52)	40.99 (11.97)	0.278
Education	16.37 (3.5)	16.5 (2.66)	0.785
Work Hours	36.2 (9.63)	34.06 (7.58)	0.093
Body Mass Index (BMI)	27.32 (4.98)	26.93 (6.02)	0.666
Trier Inventory for Chronic Stress (TICS) Total Score	41.4 (15.34)	42.55 (15.06)	0.637

**Table 2 brainsci-10-00432-t002:** Repeated measures factorial ANCOVA comparing the number of words recalled at the immediate and delayed recall *Work-Stress Memory Task* (WSMT), for each word type, by sex.

Effects	F	*df*	*p*
Phase	8.143	(1, 396)	0.005 **
Type	0.244	(2, 396)	0.784
Sex	24.618	(1, 198)	<0.001 ***
Phase × Type	0.593	(2, 396)	0.553
Phase × Sex	3.599	(1, 396)	0.059
Type × Sex	3.874	(2, 396)	0.022 *
Phase × Type × Sex	0.531	(2, 396)	0.588

* *p* < 0.05; ** *p* < 0.01; *** *p* < 0.001.

**Table 3 brainsci-10-00432-t003:** Pearson Correlation coefficients between recall of different type of words and reactive/diurnal cortisol measures.

Recall Moment	Type of Word	Reactive Cortisol Tsst (auci)				Basal Cortisol (aucg)			
		Men		Women		Men		Women	
		r	*p*	r	*p*	r	*p*	r	*p*
Immediate Recall	Work-Stress	−0.103	0.435	0.015	0.861	−0.060	0.675	−0.069	0.427
	Positive	−0.080	0.542	0.046	0.591	0.046	0.747	−0.122	0.161
	Neutral	−0.081	0.538	0.018	0.828	−0.035	0.803	−0.052	0.554
Delayed Recall	Work-Stress	−0.175	0.181	0.056	0.513	0.111	0.432	0.042	0.630
	Positive	−0.133	0.313	−0.019	0.824	0.245	0.080	−0.119	0.169
	Neutral	−0.075	0.571	−0.014	0.865	−0.035	0.807	−0.009	0.917

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
