# Peer review of "Sex Differences in Work-Stress Memory Bias and Stress Hormones"

_brainsci, 2020, doi:10.3390/brainsci10070432_

Round 1
Reviewer 1 Report
In this manuscript, Dumont et al propose a new model called work-stress memory task to assess how work-related stress affect men and women. Next, they studied whether saliva cortisol levels can be used as a biomarker for memory bias in this model. Overall, the manuscript is well written and organized. Methods and results are sufficiently described and conclusions are based on the observed effects Although WSMT was unable to detect sex differences in work-stress memory bias, authors did an excellent job in discussing probable reasons like the effect of word order and pre-existing work-place stress. There are some minor error messages in the text that should be addressed
Line 233, 247, 276: error message text.
Author Response
Reviewer 1 - Comments
R1C1 – In this manuscript, Dumont et al propose a new model called work-stress memory task to assess how work-related stress affect men and women. Next, they studied whether saliva cortisol levels can be used as a biomarker for memory bias in this model. Overall, the manuscript is well written and organized. Methods and results are sufficiently described and conclusions are based on the observed effects Although WSMT was unable to detect sex differences in work-stress memory bias, authors did an excellent job in discussing probable reasons like the effect of word order and pre-existing work-place stress.
We thank Reviewer 1 for their kind words regarding our work and for recognizing the importance of publishing mostly negative results.
R1C2 – There are some minor error messages in the text that should be addressed Line 233, 247, 276: error message text.
Thank you for pointing these errors out. While these error messages did not show up on our version of the document, they coincide with moments in the text where a dynamic reference to a Table was made. These dynamic references were removed and replaced with static text, which should exclude the possibility of having an error message.
Reviewer 2 Report
The present study examines sex differences in work-stress memory bias and in stress hormones. Data were collected from a sample of 201 employees.The study focuses on a noteworthy topic and addresses interesting questions that contribute to existing literature. The introduction is well written and mostly based on sufficient theoretical background. The analyses are appropriate to the goals of the study and the data. The results are interesting and summarized accurately. Major weaknesses of the study can be found in the Method section (e.g. inconsistencies, missing information). Also, the operationalization used is something to be queried. Finally, the findings are discussed concisely, while limitations and suggestions for future research are also mentioned. However, in the discussion section, some theoretical explanations are missing.
I thus have some revisions/considerations offered in hopes of strengthening the current paper:
Introduction:
Page 2, last paragraph: What is the rationale behind the statement, that differences in attention bias towards stress-related information may be sex-specific? Please provide more theoretical background on this assumption.
Materials and Methods:
Page 3: Please provide information about the recruitment process of the 295 participants in the original studies of 2011-2015.
Page 4: the authors mention that 201 of originally 204 working adults were included in the current set of analyses. Please specify, what "successfully completed the WSMT" means.
Page 4: Please provide information on the sample size of female and male participants respectively.
Page 4: "socio-demographic and psychosocial information": Please substantiate why the BMI and the self-reported chronic stress was assessed. What is the theoretical background behind it?
Page 4: "Work-stress memory task - WSMT": It would be beneficial to know the professional background of the 208 working adults when developing the work-stress words.
Page 4: "Trier Social Stress Task (TSST)". The current investigation examines whether male and female workers present a memory bias towards work-stress related information, and whether this bias is associated with concentrations of stress hormones in reactivity to a laboratory stressor. However, it should be considered, that the TSST is not specifically inducing "work stress" but represents a more general psychosocial stress task. Is it possible that the nature of the TSST might have influenced the results?
Page 4: "Protocol": please specify, what questionnaires were completed after the TSST.
Results:
Page 5: In the Preliminary Analysis the reference of Table 1 is missing.
Page 6: In Performance on the Work-Stress Memory Task the reference of Table 2 is missing.
Page 7:In WSMT Performance and Cortisol Measures the reference of Table 3 is missing.
Discussion:
I missed a in-depth discussion of and possible explanation for the finding that women recalled significantly more positive and neutral words than men in the current study.
Author Response
Reviewer 2 - Comments
R2C1 – The present study examines sex differences in work-stress memory bias and in stress hormones. Data were collected from a sample of 201 employees. The study focuses on a noteworthy topic and addresses interesting questions that contribute to existing literature. The introduction is well written and mostly based on sufficient theoretical background. The analyses are appropriate to the goals of the study and the data. The results are interesting and summarized accurately. Major weaknesses of the study can be found in the Method section (e.g. inconsistencies, missing information). Also, the operationalization used is something to be queried. Finally, the findings are discussed concisely, while limitations and suggestions for future research are also mentioned. However, in the discussion section, some theoretical explanations are missing.
We thank Reviewer 2 for their helpful comments. The precisions and additions made following this review will facilitate replication of this study and, for the reader, it will be easier to integrate our findings with previous literature.
I thus have some revisions/considerations offered in hopes of strengthening the current paper:
Introduction:
R2C2 – Page 2, last paragraph: What is the rationale behind the statement, that differences in attention bias towards stress-related information may be sex-specific? Please provide more theoretical background on this assumption.
In light of this comment, we believe that our explanation of cognition as a potential mechanism explaining the discrepancies between men and women in mental health issues at work needed clarification. Lines 88 to 92 have been reworked to make the theoretical background behind this idea more explicit.
“One of the mechanisms explaining the discrepancy in prevalence of work-related mental health disorders could be differences in cognitive processing of information as being stressful or threatening in men and women. More precisely, differences in attentional bias towards stress-related information may be sex-specific, and this could drive sex differences currently observed in stress responses.”
Materials and Methods:
R2C3 – Page 3: Please provide information about the recruitment process of the 295 participants in the original studies of 2011-2015.
More information about the recruitment process and the composition of the sample were added from line 142 to 148.
“The IUSMM is the largest psychiatric hospital in the province of Québec and, at the time, had a total of 1546 employees (65% women) from different professions. Participants in the study came from clinical services (29.9%), administration (17.2%), research (13.7%), social integration (11.3%), professional services (9.8%), maintenance (10.8%), general direction (4.4%) and human resources (3.0%), and were allowed to participate in the study during their normal work hours.”
R2C4 – Page 4: the authors mention that 201 of originally 204 working adults were included in the current set of analyses. Please specify, what "successfully completed the WSMT" means.
This particular sentence formulation is indeed misleading. We changed it to reflect the cause of the unsuccessful completion, which was technological. There were technical issues with the WSMT for 3 participants which made it impossible to complete.
Line 150 to 152 – “(those who successfully completed the WSMT)” / “since the WSMT could not be completed for technical reasons for 3 of the 204 participants.”
R2C5 – Page 4: Please provide information on the sample size of female and male participants respectively.
We added the number of women next to the percentage value previously included for added clarity about the composition of our sample in line 152.
R2C5 – Page 4: "socio-demographic and psychosocial information": Please substantiate why the BMI and the self-reported chronic stress was assessed. What is the theoretical background behind it?
Individuals with high BMI have substantially different cortisol levels than individuals with a BMI in medically “normal” range. A reference was added along with a short explanation of the theoretical background. As for chronic stress levels, they were assessed to examine whether men and women differed at a subjective level – the lack of group differences on this variable gives further support that any effect observed would not be due to difference in subjective reports. However, there are no formal norms to this questionnaire so it is impossible to compare our sample to the population.
Line 162 to 164: “Potential differences in socio-demographic information, BMI [65] and self-reported chronic stress [66] should be verified in order to make sure that the men and women sub-samples are comparable.”
R2C6 – Page 4: "Work-stress memory task - WSMT": It would be beneficial to know the professional background of the 208 working adults when developing the work-stress words.
These individuals attended stress management workshops and conferences developed by our laboratory and held in various workplaces. None of them were also participants in the main study. Precision about this was added.
Line 176 to 180: “In order to develop the list of work-stress words, we asked 208 working adults attending stress management workshops and conferences in different organizations to write down words that “best described stress in the workplace for them”. Note that none of the participants in the original sample of this study were part of this initiative. Individuals were mostly working in IT/Engineering (30.8%), management or direction (30.8%), administrative support (8.2%) and human resources (7.2%).”
R2C7 – Page 4: "Trier Social Stress Task (TSST)". The current investigation examines whether male and female workers present a memory bias towards work-stress related information, and whether this bias is associated with concentrations of stress hormones in reactivity to a laboratory stressor. However, it should be considered, that the TSST is not specifically inducing "work stress" but represents a more general psychosocial stress task. Is it possible that the nature of the TSST might have influenced the results?
Reviewer 2 raises an interesting point regarding the specificity of the TSST. The TSST was chosen as a psychosocial stressor given its widespread use and thorough validation. While there could be advantages to an adapted task for our research objectives, we believe including the TSST was the right methodological choice for two reasons. First, it allows comparability with the rest of the literature and second, it has been shown to reliably activate the HPA axis. It is also important to note that the first part of the TSST asks participants to do an oral presentation in the context of a job interview, which could be considered to be work-related. To address this comment, we emphasized the goal of this procedure, which is to provoke the activation of the HPA axis and added a mention of this point in the discussion of the limitations of this paper.
Line 191-192: “(…) and is one of the standard stress interventions used in a laboratory setting aimed at provoking activation of the HPA axis.”
Line 343-345: “Also, while the TSST is a validated method to induce the activation of the HPS axis, using a stressor more closely related to situational stress as it is experienced at work could have helped prime memory biases specific to work.”
R2C8 – Page 4: "Protocol": please specify, what questionnaires were completed after the TSST.
The full list of questionnaires has been added in appendix, and the questionnaire that were used in this particular paper were highlighted in the “2.3 Protocol” section.
Line 207-211: “In this paper, results to the Trier Inventory for Chronic stress will be reported, as previously mentioned. The full list of questionnaires in the original protocol can be found in appendix Table A2. Note that questionnaire instructions and practice were provided to familiarize participants with our platform; however, participants completed questionnaires at home at their leisure.”
Results:
R2C9 – Page 5: In the Preliminary Analysis the reference of Table 1 is missing.
While there was a reference to Table 1 at line 243, there were no references to Figure 1. We assume this is the issue that Reviewer 2 was bringing forward. It may also have been a similar issue to what Reviewer 1 experienced, where dynamic references to Tables and Figures seemed broken. These dynamic references have all been changed to static ones. We also added a static reference to Figure 1 at line 251. We apologize for these errors and omissions.
R2C10 – Page 6: In Performance on the Work-Stress Memory Task the reference of Table 2 is missing.
Dynamic VS static references may have been the issue here. A static reference to Table 2 is now included at line 270.
R2C11 – Page 7: In WSMT Performance and Cortisol Measures the reference of Table 3 is missing.
We believe the issue was the same as the previous comment and changes were made on line 300.
Discussion:
R2C12 – I missed a in-depth discussion of and possible explanation for the finding that women recalled significantly more positive and neutral words than men in the current study.
As this difference was not expected, we indeed remained very conservative in the interpretation of this result. We feel that this issue was covered, at least in part, between lines 374 and 381 – where we propose that a better recall of these words could be related to the different cortisol response to the TSST. However, the lack of explicit correlation between recall and cortisol during the TSST do not support this hypothesis.
We acknowledge the fact that Reviewer 2 seems to be interested in a more in-depth analysis of this result, which led us to clarify this section and highlight our reservations on the theoretical significance of this result before further replication. We also mention other studies showing altered processing of positive information in the context of mental health disorders, which feed the discussion of this point (line 356-358). To deepen our discussion, we now mention these studies again at the end of the paragraph to emphasize the next steps to test this interesting new result.
Line 381-385: “Also, as previously mentioned, some studies have found differences in processing of positive information in individuals, women in particular, with previous episodes of depression [69–71]. However, we re-iterate the necessity for replication of these results before making stronger theoretical or practical assumptions about this unexpected result.”
We thank Reviewer 2 for their significant contribution to the quality of this paper.
Round 2
Reviewer 2 Report
The manuscript has been significantly improved. I have no further comments or suggestions for the authors.